



# Aggravated Air Pollution and Health Burden due to Traffic Congestion in Urban China

Peng Wang[1,2#], Ruhan Zhang[3#], Shida Sun[4], Meng Gao[5], Bo Zheng[6], Dan Zhang[7,8], Yanli Zhang[7,*], Gregory R. Carmichael[9], and Hongliang Zhang[2,3,10*]

[1]Department of Atmospheric and Oceanic Sciences, Fudan University, Shanghai 200438, China
[2]IRDR ICoE on Risk Interconnectivity and Governance on Weather/Climate Extremes Impact and Public Health, Fudan University, Shanghai 200438, China
[3]Department of Environmental Science and Engineering, Fudan University, Shanghai 200438, China
[4]Ministry of Education Key Laboratory for Earth System Modelling, Department of Earth System Science, Tsinghua
University, Beijing 100084, China
[5]Department of Geography, State Key Laboratory of Environmental and Biological Analysis, Hong Kong Baptist University, Hong Kong SAR, 999077, China
[6]Institute of Environment and Ecology, Tsinghua Shenzhen International Graduate School, Tsinghua University, Shenzhen 518055, China
[7]State Key Laboratory of Organic Geochemistry and Guangdong Key Laboratory of Environmental Protection and Resources Utilization, Guangzhou Institute of Geochemistry, Chinese Academy of Sciences, Guangzhou 510640, China
[8]University of Chinese Academy of Sciences, Beijing,100049, China
[9]Department of Chemical and Biochemical Engineering, The University of Iowa, Iowa City, IA52242, USA
[10]Institute of Eco-Chongming (IEC), Shanghai, 202162, China

#These authors contributed equally to this work.

*Correspondence to:* Yanli Zhang (zhang_yl86@gig.ac.cn) & Hongliang Zhang (zhanghl@fudan.edu.cn)

**Abstract.** Vehicle emission is regarded as a primary contributor to air pollution and related adverse health impacts. Heavy traffic congestion increases traffic flow and thus produces more O3 precursors emissions, leading to more

adverse air quality issues. Although the development of vehicle emission inventory has received great concern and continuous efforts, limitations still exist. For example, real-time diurnal variations and increases in emission rates due to traffic congestion are not well understood. In this study, we developed a new temporal-allocation approach in transportation emission to investigate its impacts on air quality and health burden due to traffic congestion in China in 2020. Both real-time congestion level data and emission correction factors were considered in the approach.

Results show that traffic congestion aggravates air pollution and health burden across China, especially in the urban clusters such as the North China Plain and Sichuan Basin. In these regions, the average annual increases of fine particulate matter ($PM_{2.5}$) and ozone ($O_3$) could be up to 3.5 µg m$^{-3}$ and 1.1 ppb, respectively. The excess $PM_{2.5}$ and $O_3$ attributed to the traffic congestion also induce an additional 20,000 and 5,000 premature mortality in China, respectively. In major cities, the increased rate of premature mortality caused by traffic congestion may reach

17.5%. Therefore, more effective and comprehensive vehicle emission control policies or better planning of road network should be established to reduce traffic congestion and improve air quality in China.





**Graphic abstract**

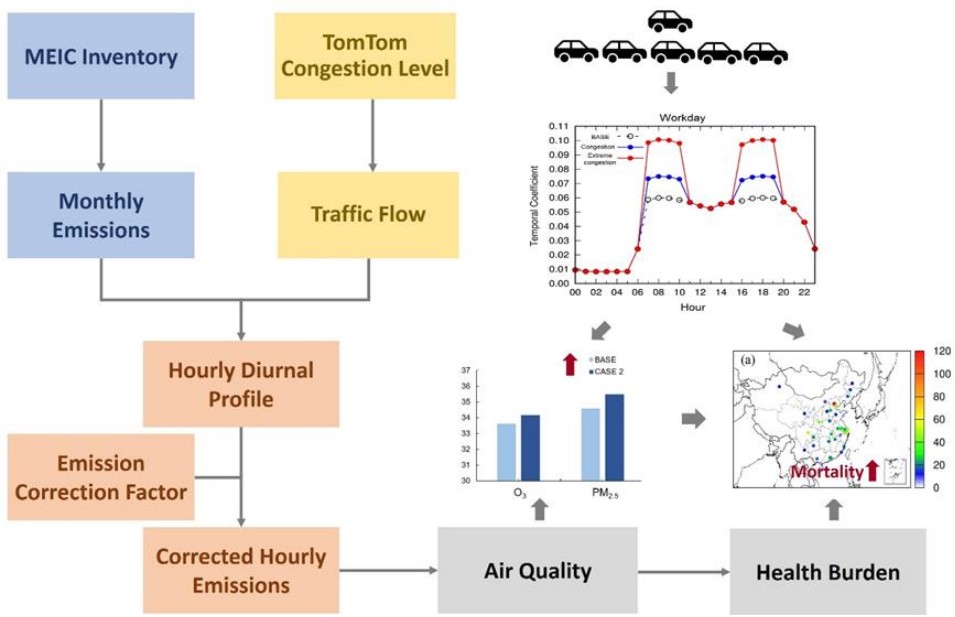





## 1 Introduction

With the rapid development of the economy and automobile industry, vehicle number has increased significantly in China in the recent decade. In June 2021, according to the Chinese government, China's vehicle number reached 384 million with the highest historical growth rate of 32.33% (compared to the year 2020)(Xinhua, 2021). However, the increasing vehicle number has deteriorated the air quality in China(Hao et al., 2007; Zhang et al., 2016; Miao et al., 2019). Xu et al. (2019) reported that vehicle volume was the most significant contributor to air pollution compared to

other factors such as population density during 2005-2016 in China. The vehicle emissions, including nitrogen oxides ($NO_x = NO_2 + NO$) and volatile organic compounds (VOCs), are the essential precursors of fine particulate matter ($PM_{2.5}$) and ozone ($O_3$)(Wang et al., 2019; Jeong et al., 2019; Li et al., 2016; Liu et al., 2017a; Yao et al., 2015). Although the Ministry of Ecology and Environment of the People's Republic of China (MEE) has implemented a series of strategies (such as updating vehicular emission standards) to reduce vehicle emissions in recent years, it was

still the dominant contributor to $PM_{2.5}$ concentration in the key regions such as Beijing-Tianjin-Hebei (BTH) in China(Gao et al., 2018). Thus, it is vital to have a comprehensive understanding of vehicle emissions, aiming to effectively alleviate the air pollution in China.

Vehicle emission also induces adverse health impacts since it is a major source of $PM_{2.5}$ and $O_3$(Levy et al., 2010;

Zhang and Batterman, 2013; Zhang et al., 2017; Shindell et al., 2011; Huang et al., 2020). Although $PM_{2.5}$ has decreased substantially in China(Zhang et al., 2019), it continues to receive substantial attention due to the strong correlation between adverse health impacts and climate change(De Kok et al., 2006; Chen et al., 2018; Xu et al., 2017; Bond et al., 2013). According to the MEE, the annual $PM_{2.5}$ concentration in China was 33 µg m$^{-3}$ in 2020, which is still ~7 times the latest World Health Organization (WHO) standard(World Health, 2021). Tong et al. (2020a) stated

that vehicle emissions have the highest impact on public health during the morning rush hours with the annual premature deaths up to 4435 (95% confidence interval (CI): 3655, 4904) in Beijing, China. Considering $O_3$, Cohen et al. (2017) reported that $O_3$ concentrations in all major Chinese metropolitan regions were at least 10% higher than the Chinese Ambient Air Quality Standard (CAAQS) level (160 µg/m$^3$). In 2015, 47,000 (CI 95: 32,000 to 70,000) fewer deaths attributable to $O_3$ exposure were projected by the implementation of vehicle emission controls(Wang et al.,

2020). In China, the transportation attributable deaths related $PM_{2.5}$ and $O_3$ in 2015 is 11% of all sources of emission(Anenberg et al., 2019). Therefore, it is significant to figure out the related health impacts of vehicle emissions to reduce premature deaths in China.

Chemical Transport Models (CTMs) have been widely used to study vehicle emissions and their impacts on air quality

(Liu et al., 2010; Che et al., 2011; Zhang et al., 2020a). Zhang et al. (2012) found that the transportation sector was an important contributor to nitrate (a major component of $PM_{2.5}$) in China by using the source-oriented version of the Community Multiscale Air Quality model (CMAQ). However, the CTM performance highly depends on the emission inventories(Hu et al., 2016a), which may lead to uncertainties in understanding vehicle emissions. In China, the vehicle





emission inventory has been developed in national, regional(Deng et al., 2020; Jiang et al., 2020), provincial(Liu et
al., 2022; Liu et al., 2017b), and city levels in China (Sun et al., 2020b; Yang et al., 2019), which is essential for
determining air pollution sources and making environmental control policies. The accurate temporal allocation of
vehicle emission inventory is beneficial for air quality simulation. Zheng et al. (2014) first calculated the monthly
vehicle emissions in China by estimating the monthly emission factors at the county-level. Sun et al. (2020a)
introduced the speed correction curves to improve the simulation of vehicle emission factors. However, there are still
shortcomings with temporal allocation in the development of vehicle emission inventories. First, most of the emission
inventories are distributed at a monthly level(Jiang et al., 2020), without providing a robust diurnal distribution profile.
Second, the emission inventory could not be updated in time and is usually available after several years of latency,
offering limited help to understand the current air pollution(Zheng et al., 2021b). Third, the changes in emission rates
due to traffic congestion are not considered(Liu et al., 2022), which could not accurately reflect the temporal emission
distribution in the inventory. In China, more than 80% of cities have suffered from heavy traffic congestion that leads
to substantial changes in air pollutants such as $PM_{2.5}$ and $O_3$(Tong et al., 2020b; Zhang et al., 2018). Consequently, a
more comprehensive temporal distribution approach of vehicle emission is urgently required.

In this study, we used real-time traffic congestion data and the updated CMAQ model (Ying et al., 2015) to investigate
the characteristics of vehicle emissions in China in 2020. The air pollution related premature death mortality (from $O_3$
and $PM_{2.5}$) was also evaluated to determine health impacts attributed to the changes in vehicle emissions. The purposes
of this study are: (1) to provide a diurnal profile for vehicle emission; (2) to improve the hourly vehicle emissions
rates based on the real-time traffic congestion data; (3) to determine the response of air quality and the associated
health impacts from the updated emissions. This study aims to give an in-depth investigation of traffic congestion and
its related air quality and health impacts, which has important implications for establishing effective control strategies
in China.

## 2 Methods

### 2.1 Temporal-allocation approach of vehicle emissions

The hourly temporal coefficient in the diurnal profile was estimated considering both the traffic flow and the emission
rate. First, the traffic flow at the city level was calculated based on TomTom congestion level data collected from
(https://www.tomtom.com/en_gb/traffic-index/ranking/, last access: 15 Sep. 2021). The TomTom congestion level
(CL) describes the extra travel time as a percentage compared to the non-congestion situation, which was obtained in
22 major cities in China (**Figure S1**). When CL is zero, the traffic is smooth without congestion, but with cars and
emissions. In these 22 cities, traffic flows increased with the severity of CL. In this study, daily and hourly congestion
105 level data were collected to achieve high temporal resolution. Then, CL was converted to traffic flow using a sigmoid
function-- Eq. (1) from(Liu et al., 2020):

$$Q = a + \frac{b \cdot CL^c}{d^c + CL^c} \tag{1}$$





where Q is the daily mean car counts, while $a, b, c,$ and $d$ are empirical parameters to fit the sigmoid function without physical meanings, and their values are 100.87, 671.06, 1.98, and 6.49, respectively, according to Liu et al. (2021).

In general, the vehicle emissions were proportional to the traffic flow (Gong et al., 2017), and the temporal coefficient was calculated as the Eq. (2):

$$HTC_{w,h} = \frac{Q_{w,h}}{\sum_h Q_{w,h}} \qquad (2)$$

where $HTC_{w,h}$ is the hourly temporal coefficient (unit: %); $w$ means weekday or weekend, for whom different traffic flows are considered separately.

Vehicle emissions were influenced by both traffic flow and emission rate(Zhang et al., 2018). During off-peak traffic hours, the emission rates were significantly lower than peak hours because vehicles were more polluted under congested conditions due to the frequent low and idle speed(Zhang et al., 2020b). To reflect the impact of congestion, this study used the temporal coefficient with the emission correction factor to reflect the emission changes in the peak hours as shown in Eq. (3):

$$E'_{w,ph} = E_m \times HTC_{w,ph} \times ECF_h \qquad (3)$$

where $E'_{w,ph}$ stands for the emission rate at peak hours ($ph$ ranges from 0:00 to 23:00); $E_m$ is the original emissions in month $m$; $ECF_h$ is the emission correction factor, determined by the driving speed from the national technical guidelines on emission inventory(Mee, 2014) (**Table S1**). ECF values were from the gasoline vehicle since it was the most dominant vehicle type in China(Wu et al., 2017).

## 2.2 CMAQ model application and validation

The CMAQ version 5.0.1 with the updated secondary organic aerosol (SOA) formation mechanism was applied in this study(Ying et al., 2015). The simulation period was the whole year 2020. The Weather Research and Forecasting (WRF) model version 4.2.1 was used to generate the meteorological inputs, using the high-resolution final (FNL) reanalysis data from the National Centers for Environmental Prediction (NCEP; https://rda.ucar.edu/datasets/ds083.3/, last access: 02 Oct. 2021). The anthropogenic emissions were from the Multi-resolution Emission Inventory for China (MEIC; http://www.meicmodel.org/, last access: 02 Oct. 2021) based on the year 2020(Zheng et al., 2021a; Zheng et al., 2018). Due to the lack of explicit vehicle emissions in the MEIC inventory, the vehicle emissions in this study were estimated based on the on-road emission ratios of the Emissions Database for Global Atmospheric Research (EDGAR; https://edgar.jrc.ec.europa.eu/; last access: Oct., 2021)(Crippa et al., 2020). The open burning and biogenic emissions were from the Fire INventory from NCAR (FINN)(Wiedinmyer et al., 2011), and the Model of Emissions of Gases and Aerosols from Nature version 2.1 (MEGAN2.1)(Guenther et al., 2012), respectively.

We set up three CMAQ simulation cases using different transportation emissions as summarized below: (1) the diurnal profile was determined using TomTom data, without considering the changes in emission rates due to congestion (BASE); (2) the diurnal profile was the same as the BASE case, and emissions rates were adjusted using the actual speed correction coefficient from **Table S1** (CASE 1); and (3) the diurnal profile was the same as the BASE case, and emissions rates were adjusted under the extreme congestion condition (emission correction factors from < 20 km h$^{-1}$





in **Table S1**, CASE 2). For CASE 1, the average speeds were 26.0 and 27.3 km h$^{-1}$ for workdays and weekends,
respectively. As a result, the emission correction factors from 20-30 km h$^{-1}$ range were used in CASE 1.

The WRF model performance is shown in **Table S2** with observation data from the National Climate Data Center
(NCDC; https://www.ncdc.noaa.gov/, last access: 20 March 2021). Four key parameters including temperature at 2m
(T2), wind speed and wind direction at 10m (WS and WD), and relative humidity (RH) were selected in the validation.
The WRF model simulated the higher T2 in the winter but lowered T2 in other seasons, indicating the mean bias (MB)
values variations. WS was slightly overpredicted for the whole year, and its gross error (GE) values all met the
benchmark(Emery et al., 2001). For WD, its MB values have met the benchmark, while the GE values were 30%
larger than the benchmark. RH was slightly overpredicted in all months. Our WRF model performance was
comparable to previous studies in China(Hu et al., 2016a), which could provide reasonable meteorological inputs for
the CMAQ model.

The CMAQ model performance of the BASE case is shown in **Table S3**. The observation data is from China National
Environmental Monitoring Centre (CNEMC; http://www.cnemc.cn/, last access: 15 Aug. 2021), and a total of 1600
sites are included in the validation. The CMAQ model predictions agrees well with the observation (except the O$_3$ in
February slightly over the criteria), which is comparable and even better than previous studies(Hu et al., 2016b; Liu
et al., 2020). Consequently, the CMAQ model provides robust results to investigate the impacts on air quality and
public health from traffic congestion.

### 2.3 Estimation of premature mortality from air pollution

The premature mortalities for PM$_{2.5}$-related and O$_3$-related diseases were estimated according to the methods as
follows.

### 2.3.1 Estimation of PM$_{2.5}$-related premature mortality

The annual premature mortalities due to long-term exposure of PM$_{2.5}$ from chronic obstructive pulmonary disease
(COPD), ischaemic heart disease (IHD), lung cancer (LC), and cerebrovascular disease (CEVD) were estimated in
this study. The relative risk (RR) from Burnett et al. (2014) was used to estimate premature mortality, as shown in
Eqs. (4) and (5):

$$RR = 1, \ for \ c < c_{cf}, \tag{4}$$

$$RR = 1 + \alpha \left\{ 1 - exp\left[-\gamma\left(c - c_{cf}\right)^{\delta}\right]\right\}, \ for \ c \geq c_{cf}, \tag{5}$$

Where $c$ is the predicted average annual PM$_{2.5}$ concentration from the CMAQ model, and $c_{cf}$ represents the threshold
concentration, below which there is no additional health risk; $\alpha$, $\beta$, and $\gamma$ are relevant parameters, calculated using the
Monte Carlo method (including 1000 simulations) from the Global Health Data Exchange (http://ghdx.healthdata.org/,
last access: 15 Aug. 2021) as described in Guo et al. (2018). In this study, RR was calculated for people above the age
of 30, and the premature mortality ($\Delta Mort$) was determined using the Eq. (6):


$$\Delta Mort = y_0 \left[\frac{RR-1}{RR}\right] Pop, \qquad (6)$$

where $y_0$ is the baseline mortality rate, obtained from the China Health Statistical Yearbook 2020
(https://www.yearbookchina.com/navibooklist-n3020013080-1.html, last access: 15 Aug. 2021), and $Pop$ is the
population data that is from China's Seventh Census data (http://www.stats.gov.cn/tjsj/tjgb/rkpcgb/, last access: 15
Aug. 2021) as shown in **Figure S1**.

### 2.3.2 Estimation of O₃-related premature mortality

In this study, China-specific concentration-response functions (CRF) were adapted to estimate the health impacts due
to O₃ exposure(Gu and Yim, 2016; Gu et al., 2018). The relative risk of mortalities with corresponding annual
maximum daily 8h average ozone (MDA8 O₃) concentrations were calculated using the Eq. (7):

$$RR = exp[\theta(c - c_{cf})], \qquad (7)$$

Where $\theta$ is fitted by meta-regression based on the previous epidemiological studies in China(Gu et al., 2018). $c$ and
$c_{cf}$ denote the average annual MDA8 O₃ concentration from the CMAQ model and the threshold value, below which
there is no additional risk, respectively. The threshold concentration of MDA8 O₃ was 70 μg/m³ in this study(Xie et
al., 2017). Same as PM₂.₅, the premature mortality induced by O₃ is calculated by Eq. (6), including cardiovascular
diseases (CDM), COPD, IHD, and LC.

## 3 Results and discussion

### 3.1 Diurnal temporal-allocation in the vehicle emissions

The congestion levels in urban China have clearly shown the workday and weekend patterns (**Figure S2**), which is
consistent with previous studies(Wen et al., 2020; Liu et al., 2018b). In general, the congestion levels are alleviated
during the weekends. On workdays, the average congestion level is 1.4 times that on weekends. Among the 22 cities,
the peak congestion level (54%) is found at 08:00 am Monday. In terms of temporal variations, the congestion levels
on workdays and weekends all present the bimodal patterns with different peak hours. As a result, the workday rush
hours are selected as 07:00-10:00 am and 4:00-7:00 pm, 10:00-11:00 am, and 14:00-19:00 on weekends. **Figure S3**
shows the 22-city average traffic flow ($Q$) calculated from Eq. (1) using the TomTom congestion data. Compared with
the level of congestion, the traffic flow on workdays shows a more similar result to weekends, which may
underestimate the peak traffic flow on workdays. Eq. (1) was derived merely based on data in Paris due to the real-
time traffic data limitation(Liu et al., 2020). Particularly, congestion level and traffic flow in Beijing, Shanghai,
Guangzhou, and Chengdu are shown in **Figures S4-S5**. The changes in Beijing and Shanghai are roughly the same.
In these two cities, traffic flows and congestion levels are higher in the morning than in the evening, which is in
contrast to Chengdu and Guangzhou. In addition, the traffic flow is largest in Beijing in the morning and Chengdu in
the evening, respectively. Using this equation in China may introduce additional uncertainties since each city has a
specific relationship between congestion level and traffic flow. Thus, more localized traffic flow data is required to
improve the accuracy of the vehicle emission inventory development.

**Figure 1** shows the hourly temporal coefficient in the diurnal profile of all cases (take VOC emissions as an example, other pollutants are similar). In the BASE case on workdays, the temporal coefficient in rush hours is much lower than the congestion level, indicating the BASE case may underestimate the emission rates. The cases considering changes in emissions (CASE 1 and 2) have comparable trends with the congestion level, with the temporal coefficient larger than 0.07 in rush hours. Considering the emissions changes, CASE 2 has the highest emissions rate (**Figures S6 and S7**). Compared with the BASE case, the anthropogenic emissions of $NO_x$, VOCs, and CO in CASE 2 have increased, especially in the key areas for air pollution prevention and control, such as North China Plain (NCP), Yangtze River Delta (YRD), and Sichuan Basin (SCB), where have higher vehicle numbers and population density. And these increases could lead to significant impacts on air quality and public health.

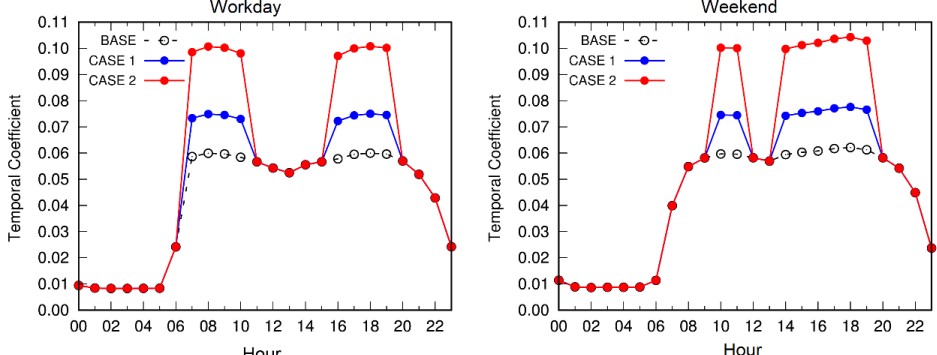

**Figure 1. The VOC temporal coefficient in the hourly diurnal profile of all simulation cases on workdays and weekends.**

**3.2 Response of air quality due to traffic congestion**

The Air Quality Index (AQI) in China is determined by the concentrations of six major pollutants: $PM_{2.5}$, $PM_{10}$, $SO_2$, $NO_2$, $O_3$, and CO. Since $PM_{2.5}$ is a major component of $PM_{10}$ and there is no obvious change in the $SO_2$ emission (**Figure S7**), the changes of other four pollutants ($PM_{2.5}$, $O_3$, $NO_2$, and CO) are discussed in this section. **Figure 2** shows the concentration and changes of these 4 pollutants due to urban traffic congestion. According to the CMAQ results, the annual average concentrations of $PM_{2.5}$, MDA8 $O_3$, $NO_x$, and CO are ~35 µg m$^{-3}$, 55 ppb, 10 ppb, and 0.36 ppm, respectively in 2020. The concentrations of the $PM_{2.5}$ and MDA8 $O_3$ are approximately 7.0 and 1.2 times of the WHO 2021 standard(World Health, 2021), which may induce severe health impacts. Ubiquitously, the peak values of these pollutants are predicted in the NCP and SCB regions, with the annual average $PM_{2.5}$ higher than 60 µg m$^{-3}$. In addition, traffic congestion has aggravated air pollution across China, which is consistent with the previous study(Xu et al., 2019). Among all simulated cases, CASE 2 (extreme congestion conditions) has the highest pollutants level. The significant enhancements in these pollutants (compared to the BASE case) are simulated in NCP and SCB. The maximum increase of $PM_{2.5}$ and $O_3$ are 3.5 µg m$^{-3}$ and 1.1 ppb, respectively.



PM$_{2.5}$, NO$_2$, and CO have lower concentrations on weekends than on workdays (**Figure S8**), similar to previous studies(Liu et al., 2018b; Wen et al., 2020; Bao et al., 2016), which is partially attributed to the lower anthropogenic emissions on weekends. Significant decreases on weekends are predicted in the NCP and the SCB regions. In the NCP region, the reduction of PM$_{2.5}$ is up to 6.0 µg m$^{-3}$. In contrast, the rising trend of O$_3$ has occurred on weekends, which is consistent with previous studies(Wang et al., 2021a; Zhao et al., 2019; Wang et al., 2021b). The elevated O$_3$ is attributed to the reduced NO$_x$ emissions on weekends, which promoted the formation of O$_3$ under a VOC-limited regime and reduced the titration impacts(Li et al., 2019; Blanchard and Tanenbaum, 2003).

To further investigate the traffic congestion impacts on air quality in urban areas, four representative mega-cities are selected (**Figure S1**): Beijing (NCP), Shanghai (YRD), Guangzhou (PRD), and Chengdu (SCB). Except for Shanghai, the traffic congestion (CASE 2) enhances the PM$_{2.5}$ concentration during the morning and evening rush hours (**Figure 3**). The peak of PM$_{2.5}$ always occurred in the morning or evening rush hours, indicating the important role of the traffic congestion in the PM$_{2.5}$ formation(Tong et al., 2020b). Different workday-weekend patterns are also found in these mega-cities. The most obvious weekend impact is in Guangzhou with more than 30% of PM$_{2.5}$ reduced in the morning peak hours (compared to workdays). Interestingly, in Shanghai the slightly rising PM$_{2.5}$ concentration is predicted on weekends, resulting from the changes of emissions and regional transport during weekends and weekdays(Atkinson-Palombo et al., 2006; Mönkkönen et al., 2004). The similar phenomenon was also reported in Nanjing(Shen et al., 2014), another megacity in the YRD region. For all these megacities, the lower NO$_2$ concentrations on weekends lead to slightly higher O$_3$ (**Figures S9-S12**).

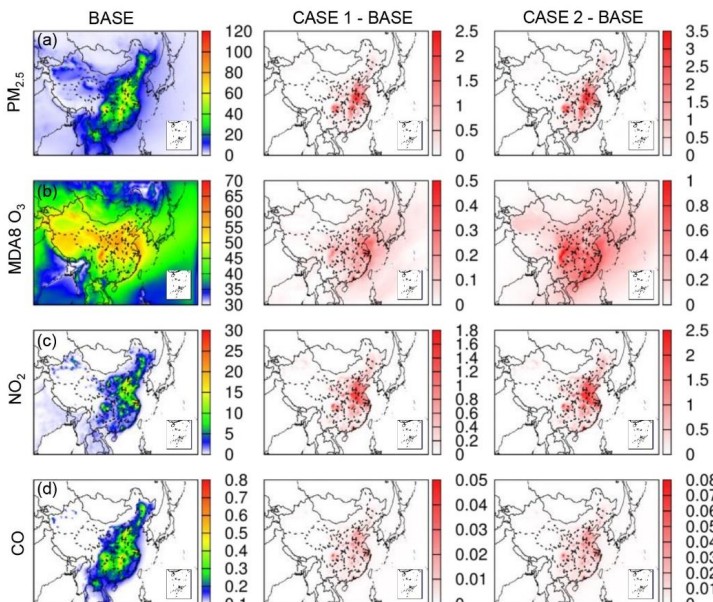

**Figure 2. The annual average concentrations of (a) PM$_{2.5}$, (b) MDA8 O$_3$, (c) NO$_2$, and (d) CO of BASE case, and their differences between CASE 1, and CASE 2, respectively, in 2020. Unit for PM$_{2.5}$ is µg m$^{-3}$, ppb for MDA8 O$_3$ and NO$_2$, and ppm for CO.**



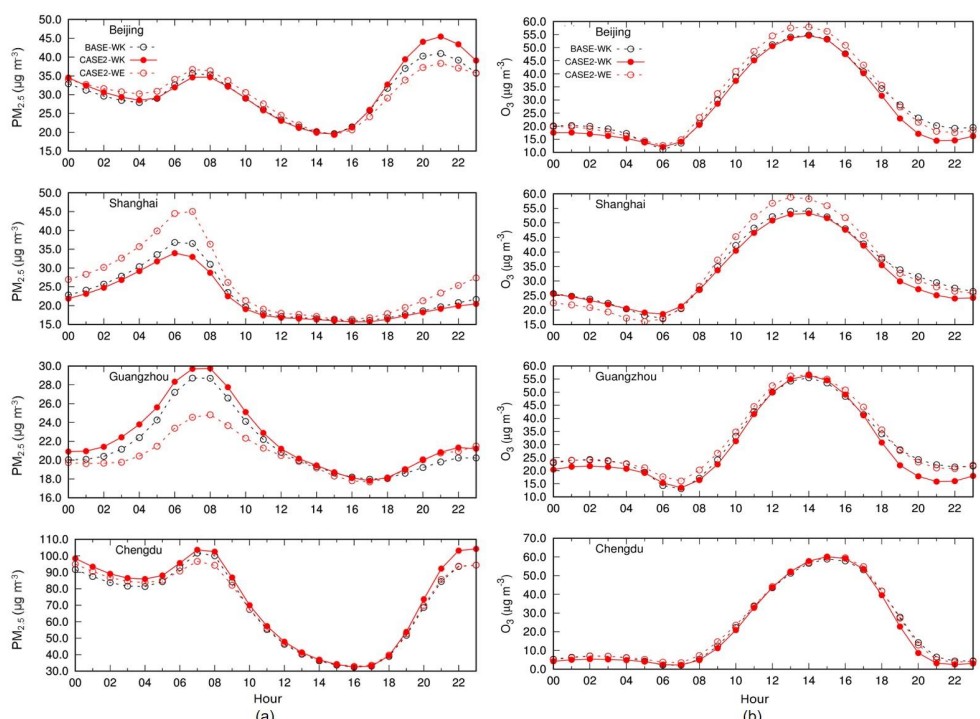


**Figure 3. The diurnal profile of PM$_{2.5}$ and O$_3$ concentrations of BASE (without considering congestion) and CASE 2 (considering congestion) in Beijing, Shanghai, Guangzhou, and Chengdu, respectively. WK: workday, and WE: weekend.**

### 3.3 The aggravated health burden due to traffic congestion

The traffic congestion leads to more severe health impacts throughout China. As shown in


**Table 1**, the total estimated PM$_{2.5}$-related annual premature mortality is 0.90, 0.91, and 0.92 million for BASE, CASE 1, and CASE 2, respectively. The extreme congestion (CASE 2) has induced an average 1.7% increase in the

total premature mortality in China. The CEVD is the most important contributor to total premature mortality, followed by IHD. With considering the extreme congestion situation (CASE 2), the CEVD and IHD cause 0.51 and 0.24 million deaths, respectively, amounting to 81% of the total premature mortality. In China, the high annual premature mortality due to excess PM$_{2.5}$ is estimated in regions with the higher PM$_{2.5}$ concentrations or population density, such as the NCP and PRD regions (**Figure 4 and Figure S13**). This result is comparable with previous

studies(Guan et al., 2019; Xie et al., 2016; Maji et al., 2018). More severe health impacts due to traffic congestion are also predicted in these regions. It is noted that in the PRD, the annual average PM$_{2.5}$ concentration is ~20 µg m$^{-3}$ (57% of the national average value), but it still experiences serious health risks, mainly attributed to the surge in population density. The population in Guangdong province (where the PRD region is located) increased by 21.7





million from 2010 to 2020, ranking first in China (http://www.stats.gov.cn/tjsj/tjgb/rkpcgb/, last access: 02 Oct.

2021). Thus, the establishment of emission control policies in the future should also refer to socioeconomic

development level besides the pollution level.

As cities are mostly affected by traffic congestion, representative cities from each province in Mainland China except
Lasa in Tibet were selected to compare the differences among regions. The most significant response to PM$_{2.5}$-related

health impacts due to the traffic congestion is in Beijing, with the additional 120 annual deaths (**Figure 4**). In Chengdu
and the major cities in the YRD (Shanghai, Suzhou, and Nanjing), traffic congestion also negatively impacts public
health. Surprisingly, in Sanya the increase of PM$_{2.5}$-related premature mortality is estimated as high as 17.5% (**Figure
S1**), which is much higher than that of most megacities. On workdays and weekends, the CEVD is the most significant
contributor to the PM$_{2.5}$-related health burden in megacities (**Figure 6**). Except for Shanghai, all mega-cities (Beijing,

Guangzhou, and Chengdu) have lower mortality on weekends, coinciding with a previous study(Tong et al., 2020b).
However, in Shanghai, a 14% increase in PM$_{2.5}$-related daily mortality on weekends is estimated due to the higher
PM$_{2.5}$ concentration.

As for O$_3$, the total estimated O$_3$-associated annual premature mortality is 0.414, 0.415, and 0.419 million for BASE,

CASE 1, and CASE 2, respectively. The extreme congestion leads to an average 1.4% increase in the total premature
mortality in China. CMD became the major disease rather than IHD. In CASE 2, with the extreme congestion situation
considered, the CEVD and IHD cause 0.26 and 0.08 million deaths, respectively, amounting to 81% of the total
premature mortality. In China, the highly annual premature mortality due to excess O$_3$ is estimated in YRD and PRD
regions (**Figure 5 and Figure S14**). This result is consistent with previous studies(Liu et al., 2018a; Chen et al., 2021).


Unlike PM$_{2.5}$, in the city level, the most significant O$_3$-related health impacts due to traffic congestion is in Chongqing,
where 139 deaths are added each year (**Figure 5**). In Chengdu, Beijing, and the major cities in the YRD (Suzhou,
Shanghai) and PRD (Foshan), negative public health impacts of excess O$_3$ are also associated with traffic congestion.
An average of 46 deaths are added in these regions as estimated in this study. In Beijing, the O$_3$-related premature

mortality increases by 1.1%. This result is comparable to Zhong (2015) where emergency ambulance call rates related
to heart disease were higher by 2.9% in Beijing, when the traffic congestion index increased by 20%. Traffic
congestion causes more O$_3$ precursors emissions to be produced, which rises O$_3$ concentration and aggravates health
risks, coinciding with previous studies(Bigazzi et al., 2015; Ahmad and Aziz, 2013; Wang et al., 2021b). Thus, more
effective vehicle control regulation in urban areas should be considered to avoid premature death from air pollution.

In addition, CDM is the major contributor to the O$_3$-related health burden on workdays and weekends. On weekends,
mortality is higher for all diseases (CDM, COPD, LC, and IHD) in these cities (Beijing, Shanghai, Guangzhou, and
Chengdu), resulting from the O$_3$ weekend effect(Zeldin et al., 1989; Tang et al., 2008) (**Figure 6**). In Beijing, mortality
on weekends is 33.6% higher than on weekdays, and 47.0% for CDM in particular.






**Table 1. Annual premature mortality (×10⁴ deaths) in China due to COPD, LC, IHD, CDM and CEVD of all simulations.**

| PM₂.₅-related | COPD | LC | IHD | CEVD | Total |
|---|---|---|---|---|---|
| BASE | 8.93 | 8.43 | 24.15 | 49.17 | 90.68 |
| CASE 1 | 9.03 | 8.54 | 24.28 | 49.74 | 91.60 |
| CASE 2 | 9.10 | 8.61 | 24.44 | 50.09 | 92.24 |
| O₃-related | COPD | LC | IHD | CDM | Total |
| BASE | 4.53 | 3.30 | 8.18 | 25.34 | 41.35 |
| CASE 1 | 4.55 | 3.31 | 8.21 | 25.47 | 41.54 |
| CASE 2 | 4.58 | 3.34 | 8.27 | 25.72 | 41.91 |

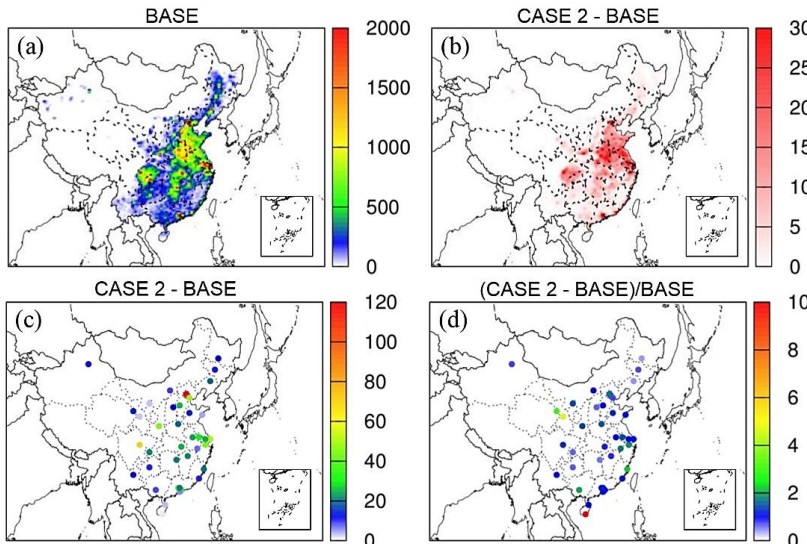

**Figure 4. The total PM₂.₅-related premature mortality of (a) BASE case, and its difference between CASE 2 across China (b) and in major cities (c) and (d).**






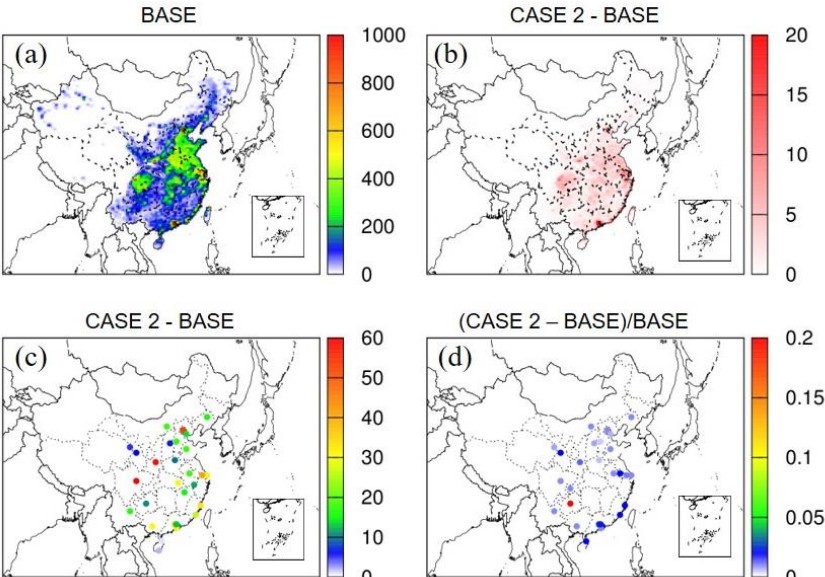

**Figure 5. The total O₃-related premature mortality of (a) BASE case, and its difference between CASE 2 across China (b) and in major cities (c) and (d).**

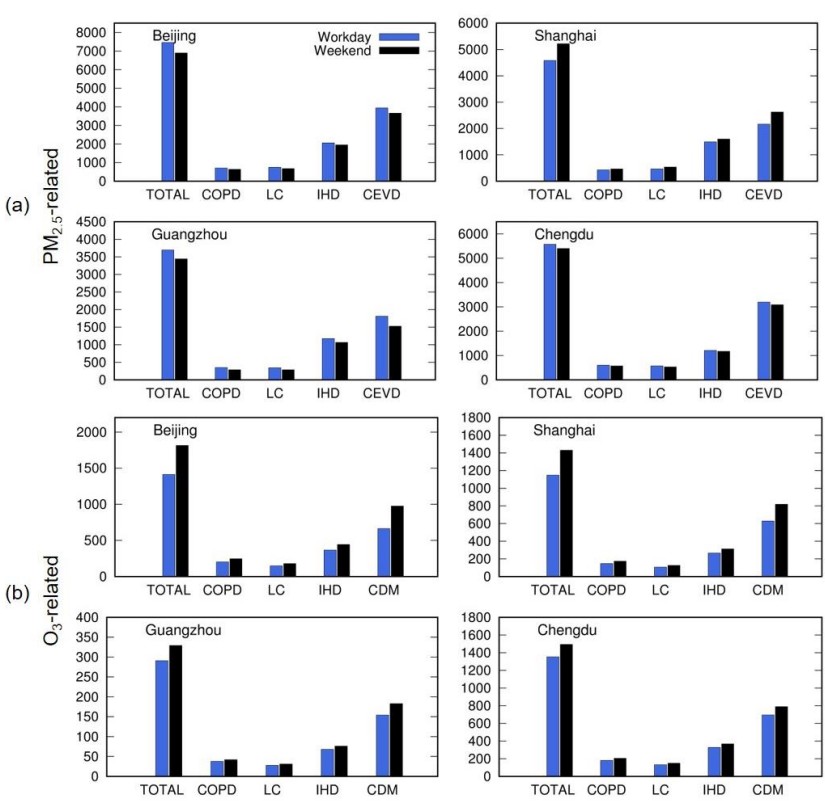


**Figure 6.** **The annual premature mortality from COPD, LC, IHD, CDM, and CEVD of CASE 2 in Beijing, Shanghai, Guangzhou, and Chengdu. (a) is PM$_{2.5}$-related and (b) is O$_3$-related.**

### 3.4 Uncertainties discussion

In this study, there are some uncertainties in the temporal-allocation approach, involving two variables: traffic flow
and emission rate. The traffic flow is calculated by using Eq. (1), which is derived based on the observation data in
Paris due to data limitations(Liu et al., 2020). Thus, it may not entirely reflect the actual traffic flow in China and
introduces uncertainties in the following CMAQ simulations. Besides, the emission rate changes only depend on the
driving speed in this study, which could cause deviation. The emission rate is influenced by various factors, such as
meteorological conditions, geographical conditions, fuel quality, deterioration level, load rating, and driving
conditions (Sun et al., 2021). However, it is challenging to incorporate all these factors into the emission rate correction
on such a national scale. Therefore, further efforts should be made to reduce the uncertainties in the vehicle emissions
inventory.



#### 4 Conclusions

In this study, we develop a new temporal-allocation approach to transportation emissions to investigate traffic congestion responses to air quality and health impacts in urban China. The real-time congestion data from TomTom is used to generate the hourly-diurnal temporal profile of the vehicle emissions, and emission correct factors are applied to qualify the emission rate changes corresponding to the traffic congestion. Our results show that traffic congestion increases pollutants concentrations, especially in the highly-developed urban clusters such as the NCP and SCB. The annual average increases of $PM_{2.5}$, MDA8 $O_3$, $NO_2$, and CO are up to 3.5 µg m$^{-3}$, 1.1 ppb, 2.5 ppb, and 0.1

ppm, respectively. In addition, the rising $PM_{2.5}$ and $O_3$ concentrations attributed to traffic congestion also enhance the health burden across China. Compared to the BASE case, the extreme congestion condition (CASE 2) induces an additional 20,000 ($PM_{2.5}$-related) and 5000 ($O_3$-related) premature mortality in China. Similar to the phenomenon in air quality, the remarkable increases are estimated in the urban clusters with higher population density. Therefore, more effective and comprehensive vehicle control policies considering socioeconomic factors should be implemented

to alleviate China's air pollution and health burden in the future.

**Data availability.** Ground-level observation data were publicly available at (https://doi.org/10.6084/m9.figshare.20015540.v1, last access: June 2022). Surface meteorological data could be found from National Climate Data Center (https://www.ncdc.noaa.gov/data-access, last access: October 2021). The Multi-resolution Emission Inventory for China could be found at (http://www.meicmodel.org, last access: October
360   2021).

**Author Contributions.** PW, YZ, and HZ designed the research. PW, SS, RZ, BZ, and DZ analyzed the data. PW and RZ performed the air quality model. PW, RZ, SS, YZ, and HZ wrote the manuscript with comments from all co-authors.

**Competing interests.** The authors declare that they have no conflict of interest.

**Disclaimer.** Publisher's note: Copernicus Publications remains neutral with regard to jurisdictional claims in published maps and institutional affiliations.

**Acknowledgments.** This work was supported by the National Natural Science Foundation of China (42077194/42061134008/42022023), the Chinese Academy of Sciences (QYZDJ-SSW-DQC032/XDPB1901), Guangdong Foundation for Program of Science and Technology Research (2020B1111360001/2020B1212060053),
Youth Innovation Promotion Association of the Chinese Academy of Sciences (Y2021096).

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
