# Peer review of "Aggravated Air Pollution and Health Burden due to Traffic Congestion in Urban China"

_Atmospheric Chemistry and Physics, 2022_

## Author Comment (AC1)

**Comments responses**

**Journal: Atmospheric Chemistry and Physics**

**Manuscript ID: acp-2022-577**

Title: "Aggravated Air Pollution and Health Burden due to Traffic Congestion in Urban China" Dear Referee #1,

We appreciate your comments to help improve the manuscript. We tried our best to address your comments and detailed responses and related changes are shown below. Our response is in blue and the modifications in the manuscript are in red.

**Referee #1**

Comments: This study focused on the impacts of traffic congestion on air quality and the associated health burden in urban areas in China. In addition, the authors also developed a new temporal-allocation approach in transportation emission to improve the current emission inventory by using the real-time congestion data. The manuscript is well written and organized, but some minor points should be improved before the publication.

Response: Thanks for the recognition of our study. Below is the response to each specific comment.

 In section 2.1, the authors used the ECF value only from the gasoline vehicles. Can the authors briefly discuss the uncertainties of this method since the diesel vehicles is also an important contributor to the transportation emission?

**Response**: Thanks for your comment. This method may slightly underestimate the vehicle emissions. Diesel vehicle is an important contributor to  $NO_x$  emission in the transportation sector (Sun et al., 2018), and our method may underestimate the  $NO_x$  emissions and the concentrations of air pollutants. However, in urban areas of China, the population of diesel vehicles is relatively small (

**Response:** Thanks for the comment. We have improved the graphic abstract accordingly.

**Reference**

Sun, S., Zhao, G., Wang, T., Jin, J., Wang, P., Lin, Y., Li, H., Ying, Q., and Mao, H.: Past and future trends of vehicle emissions in Tianjin, China, from 2000 to 2030, Atmospheric Environment, 209, 182-191, https://doi.org/10.1016/j.atmosenv.2019.04.016, 2019.

Sun, W., Shao, M., Granier, C., Liu, Y., Ye, C. S., and Zheng, J. Y.: Long-Term Trends of Anthropogenic SO2, NOx, CO, and NMVOCs Emissions in China, Earth's Future, 6, 1112-1133, https://doi.org/10.1029/2018EF000822, 2018.

Wu, Y., Zhang, S., Hao, J., Liu, H., Wu, X., Hu, J., Walsh, M. P., Wallington, T. J., Zhang, K. M., and Stevanovic, S.: On-road vehicle emissions and their control in China: A review and outlook, Science of the Total Environment, 574, 332-349, 2017.

Zhang, S., Wu, Y., Wu, X., Li, M., Ge, Y., Liang, B., Xu, Y., Zhou, Y., Liu, H., and Fu, L.: Historic and future trends of vehicle emissions in Beijing, 1998–2020: A policy assessment for the most stringent vehicle emission control program in China, Atmospheric Environment, 89, 216-229, 2014.

Zheng, B., Zhang, Q., Geng, G., Chen, C., Shi, Q., Cui, M., Lei, Y., and He, K.: Changes in China's anthropogenic emissions and air quality during the COVID-19 pandemic in 2020, Earth Syst. Sci. Data, 13, 2895-2907, 10.5194/essd-13-2895-2021, 2021.

---

## Author Comment (AC2)

**Comments responses**

**Journal: Atmospheric Chemistry and Physics**

**Manuscript ID: acp-2022-577**

**Title: "Aggravated Air Pollution and Health Burden due to Traffic Congestion in Urban China"**

Dear Referee #2,

We appreciate your comments to help improve the manuscript. We tried our best to address your comments and detailed responses and related changes are shown below. Our response is in blue and the modifications in the manuscript are in red.

**Comments:** Authors explore the potential impacts of traffic congestion on urban air quality and the corresponding health risks. This is achieved through the development of improved emission inventories by temporal-allocation of traffic congestion data. The authors utilize traffic data to correct for on-road vehicle emission rates, use the corrected emissions data to drive an air quality model, and explore the potential impacts of traffic congestion on air quality using the model generated fields. The explored concepts are interesting and provide an improved approach towards generating comprehensive emission inventories that can be used as input in air quality models, as well as informing air quality policy design and implementation. While the manuscript is clear and well structured, it can be further improved before publication by considering the following suggestions.

**Response**: Thanks for the recognition of our study. Below is the response to each specific comment.

Specific comments:

1. Please include a brief description on how (techniques/methods) TomTom estimates travel time and CL (e.g., smartphone apps) and the associated uncertainty.

**Response:** TomTom's technology takes advantage of GPS devices to determine traffic congestion levels in selected cities(Caban, 2021; Tanveer et al., 2020). With the rapid development of technology, GPS tracking technology is widely used in the traffic field to obtain people's travel patterns and real-time traffic conditions. TomTom calculates the baseline travel time by analyzing free-flow travel times of all vehicles on the entire road network – recorded 24/7, 365 days a year. The CL shows the extra time people need to spend during the congestion. The uncertainties from the TomTom may be asscociated with the

sample numbers. The TomTom data now covers 404 cities across 58 countries and most cities are highly developed, which can't represent the traffic congestion level in developing regions.

**Changes in manuscript (lines 100-104):** The TomTom data used GPS devices to estimate traffic congestion in a total of 404 cities across 58 countries, aiming to show how people were moving on the local and global levels, in real-time and over time. The TomTom congestion level (CL) describes the extra travel time as a percentage compared to the non-congestion situation, which was obtained in 22 major cities in China (Figure S1).

2. Line 104, regarding the statement "…, traffic flows increased with the severity of CL": please note in the text that according to the sigmoid relationship (Eq 1), Q asymptotes to a maximum value as CL increases.

**Response**: Thanks for your suggestion. The modifications were taken.

**Changes in manuscript:**

**Abstract (lines 104-105 ):** In these 22 cities, traffic flows asymptotes to the maximum value as CL increases due to the sigmoid relationship.

3.Line 110, "In general, the vehicle emissions were proportional to the traffic flow": please specify which emissions were proportional to traffic flow, the ones you looked at (the 2020 period) or the ones considered by Gong et al. (2017)?

**Response**: Thanks for your comments. The vehicle emissions such as carbon monoxide (CO), volatile organic compounds (VOCs), nitrogen oxide ($NO_x$), particulate matter ($PM_{2.5}$ and $PM_{10}$), and sulfur dioxide ($SO_2$) are all proportional to the traffic flow from TomTom data. The proportional method was from Gong et al. (2017).  We have revised the manuscript to clarify this information.

**Change in manuscript (lines 111-112):** In general, the vehicle emissions were proportional to the traffic flow based on TomTom data as described by  Gong et al. (2017), and the temporal coefficient was calculated as the Eq. (2):

4.Section 2.2: what were the model horizontal, vertical and temporal resolutions for CMAQ and WRF simulations. What are the potential impacts of model resolution on your results and how are your chosen resolutions justified for this study?

**Response**: Thanks for your comments. The horizontal resolutions are $36\times36$ km$^2$ and the temporal resolutions are one hour for both CMAQ and WRF models. For the vertical resolutions, a total height of ~30km divided into 18 sigma layers with the first layer height at a height of ~35 m from the surface is

applied in the WRF-CMAQ model system. The resolution plays an important role in the model's performance. If a finer horizontal resolution was used in this study, a better model performance might be expected as well as a relatively more computing time. We selected this resolution with the consideration of both model performance and computing cost. In addition, many previous studies also reported this resolution settings are able to provide robust results for the air quality analysis (Wang et al., 2021; Zhu et al., 2021; Hu et al., 2016).

5.Section 3.4: how about uncertainties in TomTom congestion level data?

**Response**: The uncertainties in TomTom congestion level data are mainly from the number of samples. Although the TomTom database covers 404 cities across 58 countries, most cities are located in high-developed regions. The congestion level in suburban and rural regions still lack. In our study, we use the TomTom data to adjust the transport emission at the national level, which may overestimate the congestion level in the suburban and rural regions.

6.Taking vehicle speed into account to correct for estimated emission rates is a step in the right direction and as you have shown has important implications for accurately forecasting urban air quality levels. Considering other relevant processes, it has been shown that Vehicle Induced Turbulence (VIT) can impact pollutant dispersion/transport in the atmosphere and it is advantages to take VIT into account for sub-grid diffusion parameterizations in air quality models (e.g., Makar et al., 2021). Please comment on whether this has been accounted for in your CMAQ simulations - also what implications can VIT have for your results.

**Response**: Thanks for your suggestion. The VIT process has not been considered in our study due to data limitations. The implementation of the VIT in the air quality model may improve the model performance and more remarkable changes in air quality will be predicted under the congestion condition. We plan to cooperate with the field measurement study to form the localized VIT parameterization in China and couple the VIT process in the future study.

Technical corrections:

1.Graphic abstract: figure sizes and figure font sizes (e.g., title, legends, labels, tick labels) can be increased to improve readability. (a) needs to be removed from the Mortality figure (bottom-right).

**Response**: Thanks for your suggestion. The modifications were taken.

[Figure]

2.Please add label (with units) to Figure S1b color-bar.

**Response**: Thanks for your suggestion. The modifications were taken.

**Changes in SI (Figure S1):**

[Figure]

**Figure S1.** (a) is the CMAQ study domain and the key cities in China, and (b) is population data used in this study. For panel (a), red circle: cities in the TomTom data base, red dot: megacities (Beijing, Shanghai, Guangzhou, and Chengdu), black circle: other key cities in China such as the provincial capital.

3.Line 117: revise "… vehicles were more polluted" to "… vehicles were more polluting"

**Response**: Thanks for your suggestion. The modifications were taken.

**Changes in manuscript: (lines 117-119):** During off-peak traffic hours, the emission rates were significantly lower than peak hours because vehicles were more polluting under congested conditions due to the frequent low and idle speed (Zhang et al., 2020).

4.Line 199 and 200: consider revising "As a result, the workday rush hours are selected as 07:00-10:00 am and 4:00-7:00 pm, 10:00-11:00 am, and 14:00-19:00 on weekends".

**Response**: Thanks for your suggestion. We have revised the sentence.

**Changes in manuscript (lines 205-207):** As a result, rush hours are selected as 07:00-10:00 am, 4:00-7:00 pm, and 10:00-11:00 am, 2:00-7:00 pm for workdays and weekends, respectively.

5.Line 219: consider revising "…, where have higher vehicle numbers and population density".

**Response**: Thanks for your suggestion. The revision was taken.

**Changes in manuscript (lines 223-225):** Notable increases in $NO_x$, VOCs, and CO emissions are found in CASE 2 compared with the BASE case, especially in the areas with higher vehicle numbers and population density, such as the North China Plain (NCP), Yangtze River Delta (YRD), and Sichuan Basin (SCB).

6. Please consider also including BASE-WE on Figure 3 and Figures S9 to S12.

**Response:** Thanks for the comments. The modifications were taken.

**Changes in manuscript(Figure 3 and Figures S9-S12):**

[Figure]

**Figure 1.** The diurnal profile of $PM_{2.5}$ and $O_3$ concentrations of BASE (without considering congestion) and CASE 2 (considering congestion) in Beijing, Shanghai, Guangzhou, and Chengdu, respectively. WK: workday, and WE: weekend.

[Figure]

**Figure S9.** The diurnal O$_3$, NO$_2$, and CO concentrations of BASE and CASE 2 in Beijing. WK: workday, and WE: weekend.

[Figure]

**Figure S10.** The diurnal O$_3$, NO$_2$, and CO concentrations of BASE and CASE 2 in Shanghai. WK: workday, and WE: weekend.

[Figure]

**Figure S11.** The diurnal O₃, NO₂, and CO concentrations of BASE and CASE 2 in Guangzhou. WK: workday, and WE: weekend.

[Figure]

**Figure S12.** The diurnal O₃, NO₂, and CO concentrations of BASE and CASE 2 in Chengdu. WK: workday, and WE: weekend.

**Reference**

Caban, J.: Traffic congestion level in 10 selected cities of Poland, Sci. J. Sil. Univ. Technol. Ser. Transp, 112, 17-31, 2021.

Gong, M., Yin, S., Gu, X., Xu, Y., Jiang, N., and Zhang, R.: Refined 2013-based vehicle emission inventory and its spatial and temporal characteristics in Zhengzhou, China, Sci Total Environ, 599-600, 1149-1159, 10.1016/j.scitotenv.2017.03.299, 2017.

Hu, J., Chen, J., Ying, Q., and Zhang, H.: One-year simulation of ozone and particulate matter in China using WRF/CMAQ modeling system, Atmospheric Chemistry and Physics, 16, 10333-10350, 2016.

Tanveer, H., Balz, T., Cigna, F., and Tapete, D.: Monitoring 2011–2020 traffic patterns in Wuhan (China) with COSMO-SkyMed SAR, amidst the 7th CISM military world games and COVID-19 outbreak, Remote Sensing, 12, 1636, 2020.

Wang, P., Shen, J., Xia, M., Sun, S., Zhang, Y., Zhang, H., and Wang, X.: Unexpected enhancement of ozone exposure and health risks during National Day in China, Atmospheric Chemistry and Physics, 21, 10347-10356, 2021.

Zhang, Y., Deng, W., Hu, Q., Wu, Z., Yang, W., Zhang, H., Wang, Z., Fang, Z., Zhu, M., Li, S., Song, W., Ding, X., and Wang, X.: Comparison between idling and cruising gasoline vehicles in primary emissions and secondary organic aerosol formation during photochemical ageing, Science of The Total Environment, 722, 137934, https://doi.org/10.1016/j.scitotenv.2020.137934, 2020.

Zhu, S., Poetzscher, J., Shen, J., Wang, S., Wang, P., and Zhang, H.: Comprehensive insights into O3 changes during the COVID-19 from O3 formation regime and atmospheric oxidation capacity, Geophysical research letters, 48, e2021GL093668, 2021.

---

## Author Comment (AC3)

**Comments responses**

**Journal: Atmospheric Chemistry and Physics**

**Manuscript ID: acp-2022-577**

**Title: "Aggravated Air Pollution and Health Burden due to Traffic Congestion in Urban China"**

Dear Referee #1,

We appreciate your comments to help improve the manuscript. We tried our best to address your comments and detailed responses and related changes are shown below. Our response is in blue and the modifications in the manuscript are in red.

Referee #1

Comments:This study focused on the impacts of traffic congestion on air quality and the associated health burden in urban areas in China. In addition, the authors also developed a new temporal-allocation approach in transportation emission to improve the current emission inventory by using the real-time congestion data. The manuscript is well written and organized, but some minor points should be improved before the publication.

**Response**: Thanks for the recognition of our study. Below is the response to each specific comment.

1.  In section 2.1, the authors used the ECF value only from the gasoline vehicles. Can the authors briefly discuss the uncertainties of this method since the diesel vehicles is also an important contributor to the transportation emission?

**Response**: Thanks for your comment. This method may slightly underestimate the vehicle emissions. Diesel vehicle is an important contributor to $NO_x$ emission in the transportation sector (Sun et al., 2018), and our method may underestimate the $NO_x$ emissions and the concentrations of air pollutants. However, in urban areas of China, the population of diesel vehicles is relatively small (< 5%) (Sun et al., 2019) due to emission control policies. For example, in Beijing, the registration of light diesel vehicles has been banned since 2003 (Zhang et al., 2014; Wu et al., 2017). Gasoline vehicle is the dominant contributor to the air quality in urban areas.

**Changes in manuscript (lines 126-128):** Our method may slightly underestimate the transportation emissions since diesel vehicle was an important contributor to $NO_x$ emissions (Sun et al., 2018), which would be improved in the future.

2. In section 2.2, the authors mentioned that the CMAQ model was with "updated SOA formation mechanism". What are the improvements of this model?

**Response:** Thanks for your suggestion. An improved SAPRC-11(S11) photochemical mechanism was incorporated into this model to treat isoprene oxidation and additional SOA formation pathways. The surface uptake of dicarbonyls and isoprene epoxides, as well as predictions of glyoxal and methylglyoxal are considered in the SOA modul. We have made corresponding modifications to introduce the SOA improvements of this model.

**Changes in manuscript (lines 131-133):** The mechanism incorporated a more explicit description of isoprene oxidation chemistry and isoprene SOA formation pathways. The surface uptake of dicarbonyls and isoprene epoxides, glyoxal, and methylglyoxal SOA formation pathways were all considered in the model.

3. Can the authors add the WRF scheme set-up to provide more details of the meteorology simulations?

**Response:** The WRF scheme set-up details are added in the supplemental information (Table S2).

**Changes in SI:**

Table S2. The WRF model set-up.

.

| Physical mechanism | Scheme |
|---|---|
| mp_physics | Thompson |
| ra_lw_physics | RRTM |
| ra_sw_physics | Goddard short wave |
| sf_surface_physics | Unified Noah |
| bl_pbl_physics | YSU |
| cu_physics | Grell-Freitas ensemble scheme |

4. The study was conducted in the 2020 when the COVID-19 happened across China. If possible, could the author simply discuss/compare the congestion impacts during the COVID-19 period and normal year (such as 2019)?

**Response:** Thanks for the comment. More obvious impacts on air quality due to traffic congestion should be found in the normal year. During the COVID-19 lockdown, on-road emissions decreased significantly due to the reduced anthropogenic activities (Zheng et al., 2021). The drastic decreases in both traffic flow (~70%) and NOx emissions (~ 40%) were reported during the COVID-19 lockdown, which may even eliminate traffic congestion.

**Changes in manuscript (lines 241-244):** Our simulation was conducted in 2020, covering the COVID-19 lockdown period. During the lockdown, a drastic decrease was reported in traffic flow (~70%), which

may even eliminate the traffic congestion and its impacts on air quality(Huang et al., 2020a; Zheng et al., 2021b). More remarkable changes in air quality associated the traffic congestion are expected during the normal year.

5.   The manuscript mentioned that the population data is from China's Seventh Census. Please explain more how to use/adjust the population data.

**Response:** Thanks for the comment. We used total population data for 2008 as the baseline and adjust the data to 2020  according to the population change rates in each province from China's Seventh Census (http://www.stats.gov.cn/tjsj/tjgb/rkpcgb/).

6.   Please improve the graphic abstract such as adding the unit of the histogram figure and deleting "(a)" in the regional plot.

**Response:** Thanks for the comment.  We have improved the graphic abstract accordingly.

[Figure]

**Reference**

Sun, S., Zhao, G., Wang, T., Jin, J., Wang, P., Lin, Y., Li, H., Ying, Q., and Mao, H.: Past and future trends of vehicle emissions in Tianjin, China, from 2000 to 2030, Atmospheric Environment, 209, 182-191, https://doi.org/10.1016/j.atmosenv.2019.04.016, 2019.

Sun, W., Shao, M., Granier, C., Liu, Y., Ye, C. S., and Zheng, J. Y.: Long-Term Trends of Anthropogenic $SO_2$, $NO_x$, CO, and NMVOCs Emissions in China, Earth's Future, 6, 1112-1133, https://doi.org/10.1029/2018EF000822, 2018.

Wu, Y., Zhang, S., Hao, J., Liu, H., Wu, X., Hu, J., Walsh, M. P., Wallington, T. J., Zhang, K. M., and Stevanovic, S.: On-road vehicle emissions and their control in China: A review and outlook, Science of the Total Environment, 574, 332-349, 2017.

Zhang, S., Wu, Y., Wu, X., Li, M., Ge, Y., Liang, B., Xu, Y., Zhou, Y., Liu, H., and Fu, L.: Historic and future trends of vehicle emissions in Beijing, 1998–2020: A policy assessment for the most stringent vehicle emission control program in China, Atmospheric Environment, 89, 216-229, 2014.

Zheng, B., Zhang, Q., Geng, G., Chen, C., Shi, Q., Cui, M., Lei, Y., and He, K.: Changes in China's anthropogenic emissions and air quality during the COVID-19 pandemic in 2020, Earth Syst. Sci. Data, 13, 2895-2907, 10.5194/essd-13-2895-2021, 2021.